# RAGE as a Novel Biomarker for Prostate Cancer: A Systematic Review and Meta-Analysis

**DOI:** 10.3390/cancers15194889

**Published:** 2023-10-09

**Authors:** Catherine C. Applegate, Michael B. Nelappana, Elaine A. Nielsen, Leszek Kalinowski, Iwona T. Dobrucki, Lawrence W. Dobrucki

**Affiliations:** 1Department of Bioengineering, University of Illinois at Urbana-Champaign, Urbana, IL 61801, USA; cca2@illinois.edu (C.C.A.);; 2Beckman Institute for Advanced Science and Technology, University of Illinois at Urbana-Champaign, Urbana, IL 61801, USA; 3Division of Medical Laboratory Diagnostics—Fahrenheit Biobank BBMRI.pl, Medical University of Gdansk, 80-211 Gdansk, Poland; 4BioTechMed Centre, Department of Mechanics of Materials and Structures, Gdansk University of Technology, 80-211 Gdansk, Poland; 5Department of Biomedical and Translational Sciences, Carle-Illinois College of Medicine, University of Illinois at Urbana-Champaign, Urbana, IL 61853, USA; 6Cancer Center at Illinois, University of Illinois at Urbana-Champaign, Urbana, IL 61801, USA

**Keywords:** receptor for advanced glycation end-products, RAGE, prostate cancer, systematic review, meta-analysis, biomarker

## Abstract

**Simple Summary:**

Prostate cancer (PCa) is a commonly diagnosed cancer among men worldwide, and the current research aims to understand the molecular factors that may help to diagnose or treat PCa. This study was conducted to evaluate all existing literature measuring the expression of the receptor for advanced glycation end-products (RAGE) in PCa using both clinical biopsies and preclinical research, performed using cell culture. Importantly, the results of this study can be used to diagnose PCa, as the results demonstrate that RAGE is more highly expressed in malignant compared to benign tissues. Additionally, the results show that RAGE activity in PCa cells activates growth, suggesting that RAGE can also be used as a therapeutic target for treating PCa. One method to reduce RAGE expression and activity that is immediately actionable is to reduce the intake of dietary advanced glycation end-products (AGEs), which are found in high levels in the Western diet.

**Abstract:**

The receptor for advanced glycation end-products (RAGE) has been implicated in driving prostate cancer (PCa) growth, aggression, and metastasis through the fueling of chronic inflammation in the tumor microenvironment. This systematic review and meta-analysis summarizes and analyzes the current clinical and preclinical data to provide insight into the relationships among RAGE levels and PCa, cancer grade, and molecular effects. A multi-database search was used to identify original clinical and preclinical research articles examining RAGE expression in PCa. After screening and review, nine clinical and six preclinical articles were included. The associations of RAGE differentiating benign prostate hyperplasia (BPH) or normal prostate from PCa and between tumor grades were estimated using odds ratios (ORs) and associated 95% confidence intervals (CI). Pooled estimates were calculated using random-effect models due to study heterogeneity. The clinical meta-analysis found that RAGE expression was highly likely to be increased in PCa when compared to BPH or normal prostate (OR: 11.3; 95% CI: 4.4–29.1) and that RAGE was overexpressed in high-grade PCa when compared to low-grade PCa (OR: 2.5; 95% CI: 1.8–3.4). In addition, meta-analysis estimates of preclinical studies performed by albatross plot generation found robustly positive associations among RAGE expression/activation and PCa growth and metastatic potential. This review demonstrates that RAGE expression is strongly tied to PCa progression and can serve as an effective diagnostic target to differentiate between healthy prostate, low-grade PCa, and high-grade PCa, with potential theragnostic applications.

## 1. Introduction

Prostate cancer (PCa) is the most commonly diagnosed cancer and the second leading cause of cancer-related deaths among men in the United States [1]. The current screening and diagnostic pathway for PCa consists of testing the serum levels of prostate-specific antigen (PSA) and performing a transrectal ultrasound-guided biopsy to histologically confirm PCa and rule out benign prostatic hyperplasia (BPH). This current standard of care has been shown to carry significant disadvantages: potential for sepsis and post-operational complications; omission of parts of the prostate due to patient pain; significant false-negative risk; and overdiagnosis due to the inability to differentiate between clinically significant and insignificant cancer cells effectively [2]. Perhaps as a result of new, more limited screening recommendations and more intensive initial treatments, diagnoses of localized disease are trending downward while the incidence of metastatic PCa is increasing [3]. To address these issues, new biomarkers, such as prostate-specific membrane antigen (PSMA), are being sought out as alternatives to PSA screening, which may provide additional information that will enable clinicians to differentiate between benign tissue, low-grade PCa, and high-grade PCa [4,5,6].

Chronic inflammation associated with increased body adiposity is a critical part of the initiation and development of PCa and other solid cancers [7]. The Western diet has been demonstrated to play a prominent role in the development of obesity and obesity-associated carcinogenesis [8]. In fact, the Western diet has been shown to exacerbate PCa tumorigenesis [9,10] and has been associated with increased mortality following PCa diagnosis [11]. Comprised largely of animal protein and high-carbohydrate and high-fat processed foods, the Western diet is also rich in dietary advanced glycation end-products (AGEs) [12]. AGEs are stable end-products formed endogenously and exogenously through the nonenzymatic glycation of proteins, lipids, and nucleic acids, which can form toxic crosslinks with other molecules and bind to specific inflammatory receptors [13]. High levels of dietary AGEs are associated with the development of inflammation-related chronic pathologies, including cancer [14,15].

The receptor for advanced glycation end-products (RAGE) is a member of the immunoglobulin protein family of cell surface proteins found in a wide range of tissue types [16]. RAGE activation by AGEs stimulates the PI3K-mediated activation of NF-κB, which leads to a positive feed-forward cascade of pro-inflammatory responses, including an increased expression of RAGE [17,18,19,20,21,22]. The activation of RAGE can also be induced by a wide range of ligands, such as the S100 protein family and high mobility group box 1 protein (HMGB1), a damage-associated molecular pattern (DAMP) molecule released by damaged cells [20]. The activation of RAGE by proteins such as HMGB1, which is released by cells that die during PCa treatment as a result of therapy such as radiation, suggests that RAGE may play a role in PCa treatment resistance. In fact, inflammation in the PCa tissue microenvironment has been linked to proliferation, apoptosis inhibition, treatment and immune resistance, angiogenesis, and epithelial–mesenchymal transition (EMT) [23]. Despite a number of studies showing the important role that the AGE/RAGE axis has been shown to play in PCa, no systematic reviews or meta-analyses evaluating RAGE expression in clinical cancer specimens exist.

Our research group has looked to RAGE as a potential target for diagnostic and therapeutic action due to their critical role in the inflammation of the PCa tissue microenvironment. We have published work investigating the use of a multimodal nanoparticle targeted at RAGE for imaging PCa, and we found that not only did RAGE allow for high-specificity imaging of PCa tissue by PET/CT, but also that RAGE expression had a strong correlation to the Gleason score [24,25]. As such, we hypothesized that RAGE would make an excellent theragnostic target in PCa; however, RAGE remains an overlooked biomarker in the context of any cancer.

In this systematic review and meta-analysis, our objectives were to explore the association between RAGE expression in clinical specimens of low- and high-grade PCa compared to normal prostate or BPH. We additionally expanded our systematic review to include preclinical works investigating the relationship between RAGE and PCa growth and markers of invasiveness and metastasis. The results of this review provide novel evidence to show that RAGE may serve as a biomarker able to not only diagnose PCa but also to differentiate among benign tissue, low-grade PCa, and high-grade PCa.

## 2. Materials and Methods

### 2.1. Study Selection Criteria

This clinical meta-analysis was conducted in accordance with the Cochrane Handbook [26] and the 2020 PRISMA guidelines for meta-analyses [27]. Studies that met the following criteria were included in the meta-analysis: (a) used validated PCa samples against appropriate control samples (normal prostate, benign prostatic hyperplasia [BPH], or prostatitis); (b) directly measured RAGE expression by validated techniques; (c) methodology was documented in replicable detail; (d) evaluated the relationship between RAGE expression and PCa; and (e) written in English.

Currently, no validated guidelines or tools exist for conducting systematic reviews and evaluating the validity and quality of mechanistic studies through meta-analyses. As part of an effort to utilize a cohesive and standardized set of guidelines for systematically reviewing and pooling evidence from preclinical studies, this systematic review was conducted in accordance with the framework outlined by the World Cancer Research Fund (WCRF) International/University of Bristol (UoB) [28]. In addition, care was taken to follow the PRISMA reporting guidelines as closely as possible [27]. Preclinical (in vitro) studies that met the following criteria were included in this systematic review: (a) examined RAGE expression in a validated PCa cell line; (b) evaluated the direct relationship between RAGE expression and PCa growth or invasion through cell culture studies utilizing validated growth or functional assays; (c) methodology was documented in replicable detail; and (d) written in English.

### 2.2. Literature Search

To reduce the risk of publication bias, we included grey literature in our search strategy. As such, we conducted a comprehensive literature search of PubMed, Web of Science, the Cochrane Library, Scopus, ProQuest, and arXiv using a combination of the following keywords and their variants: prostate cancer, prostate (adeno)carcinoma, receptor for advanced glycation end-products, advanced glycation end-product(s), N(6)-carboxymethyllysine (up to 1 August 2023). The titles and abstracts of articles identified by the keyword search were screened against the study selection criteria. Potentially relevant manuscripts were retrieved for evaluation of the full text. We also conducted a reference list search (backward search) and cited reference search (forward search) from manuscripts meeting the study selection criteria. The articles identified through this process were further screened and evaluated using the same criteria until no further relevant articles were found. Three authors (CA, MN, LN) individually determined the inclusion/exclusion of all articles retrieved in full text, and discrepancies were resolved through discussion.

### 2.3. Data Extraction and Quality Assessment

The following information was extracted from each study for the clinical meta-analysis: name of first author; year of publication; number of cases, controls, and total number of participants in the study; methodological details describing RAGE expression measurements; PCa grade, Gleason score, Gleason grade, and Gleason group; and study type. The number of cases (RAGE+) within the total numbers of both the PCa cases and controls were used to calculate the odds ratios (OR) and 95% confidence intervals (CI), which were subsequently used to perform the meta-analysis. For the secondary outcome, a grade of 3 or a Gleason score ≤ 7 was considered to be low-grade PCa, while grade 4–5 or a Gleason score ≥ 8 was considered to be high-grade PCa.

The quality assessment (QA) of each study was performed using the Newcastle–Ottawa Scale, which is a validated scale for non-randomized cohorts in a meta-analysis [29]. This tool judges the literature based on the following three categories: selection of cases and controls, comparability of studies, and exposure to the main variable (RAGE). We regarded scores of 1–3, 4–6, and 7–9 as low, moderate, and high quality, respectively. The QA scores were utilized to measure the strength of the evidence given by each study and were not used to determine the inclusion of studies.

For preclinical (in vitro) studies, data extraction was performed using the recommendations set forth by the WCRF/UoB framework as a guide [28]. The following information was extracted from each cell culture study: names of cell lines; whether cell lines were established patient-derived tumor cell lines or freshly isolated primary cells; whether cell lines were authenticated; culture conditions; treatment regime (dose and length of treatment), if any; details of laboratory procedures; RAGE-related outcomes analyzed (growth and metastatic potential); results of RAGE-related outcomes; sample size and standard deviation (SD); statistical test conducted; and *p*-values.

There is a lack of validated QA tools to evaluate the risk of bias associated with preclinical studies. As such, the QA of the cell culture studies included in this review was performed using adapted criteria recommended by the WCRF/UoB framework [28] and other published recommendations [30] (score range: 0–6; a score of 0 was assigned for each parameter not fulfilled or not reported). Based on their score, the studies were rated as low (0–3), moderate (4–6), or high (7–8) quality. The QA scores were utilized to provide a measure for the strength of the evidence and to determine if a risk of bias was present for each study and were not used to determine the inclusion of studies. Conclusions based on whether the included studies supported the biological plausibility of the causal pathway being investigated were based on the QA.

### 2.4. Statistical Analysis

STATA/IC version 14.2 (StataCorp LP, College Station, TX, USA) was utilized to analyze the data. The OR and 95% CI were used as measures of the effect size for all studies. Heterogeneity among the studies was assessed using the *I*^2^ statistic based on Cochran’s *Q* [31]. Random-effects models (DerSimonian–Laird) were applied due to high *I*^2^ values (≥50%), indicative of increased study heterogeneity. Potential publication bias was assessed using a visual assessment of funnel plots [32,33], and funnel plot asymmetry was evaluated using the Harbord test for small study effects, which is a modified version of Egger’s linear regression test [34]. We also performed sensitivity “leave-one-out” analyses to evaluate whether the pooled results differed if a single study at a time was excluded.

The extreme degree of heterogeneity between the methodologies and outcome measures of preclinical studies made conducting a true statistical analysis via a meta-analysis exceedingly difficult. In lieu of a true meta-analysis, the effect estimates for each study outcome were calculated through the generation of albatross plots. An albatross plot, as described by Harrison et al. [35], scatters the *p*-values of each study according to their sample size and according to the observed direction of the effect on the outcome (positive or negative). In the absence of exact *p*-values provided, the most conservative *p*-value was assigned to that outcome (e.g., if given *p* < 0.05, set *p* = 0.05). The contour lines extending over the plot represent the estimated effect sizes (represented as the standardized mean difference (SMD)) to allow for the estimation of the magnitude of effects for individual studies on either PCa growth or PCa metastatic potential. Visual inspection of the albatross plots determined an overall estimated standardized effect for each outcome, which represents the strength of the associations using the absolute values for the calculated beta-coefficients. As such, a larger beta-coefficient represents a larger standardized effect on that outcome. To provide additional information, meta-analyses of the *p*-values were conducted for each outcome using Fisher’s combined *p*-value. The albatross plots were generated using the package available in STATA/IC version 14.2 (StataCorp LP). A *p*-value of less than 0.05 was considered statistically significant for all analyses.

## 3. Results

### 3.1. Literature Search

In total, 198 articles evaluating RAGE in PCa were identified from the library search engines. This process is summarized in Figure 1. After removing duplicates and screening the abstracts for pertinent information, 28 articles remained and were evaluated by full-text review. The manuscripts found not to meet the inclusion criteria did not measure RAGE expression in association with the listed outcomes. Fourteen manuscripts were found to meet the inclusion criteria and were included in the final analysis: nine articles measuring the clinical expression of RAGE in PCa [24,36,37,38,39,40,41,42,43], and six articles measuring the in vitro effects of RAGE expression on PCa growth and metastatic potential [21,40,44,45,46,47]. Cell culture studies were further stratified according to the measured RAGE-dependent outcomes: five studies evaluated PCa cell proliferation [40,44,45,46,47]; and three studies [21,40,47] evaluated invasion, migration, and/or EMT markers (metastatic potential) of PCa cells.

### 3.2. Study Characteristics

All nine clinical studies [24,36,37,38,39,40,41,42,43] were retrospective case-control studies examining the incidence of RAGE expression in PCa samples vs. benign controls. The study characteristics are summarized in Table 1. The total number of study participants included was 761, and the total number of PCa cases reported was 421. All articles reported incidence data as positive vs. negative RAGE expression, enabling the calculation of the ORs and corresponding 95% CIs.

Quality assessment (QA) scores were assigned to each article using the criteria outlined by the Newcastle–Ottawa scales for case-control studies [29]. The average score for the studies was 5.8 ± 1.5, the lowest score being 3 (low quality) and the highest being 8 (high quality) on the 9-point scale. One study [38] was considered low-quality, five studies [24,36,41,42,43] were considered to be of moderate quality, and three studies [37,39,40] were considered to be high-quality. The QA scores for the individual studies are provided in the Appendix A.

The cell culture study characteristics and results are summarized in Table 2. All six cell culture studies identified used a single primary PCa tumor cell line (DU145, LNCaP, or PC-3) in their analyses. All studies quantified RAGE expression in association with the measured outcomes, and five [21,44,45,46,47] of the studies directly modulated RAGE expression to examine its effects on PCa cell growth and/or metastatic potential.

QA of the cell culture studies was carried out using criteria adapted from the WCRF/UoB framework for evaluating mechanistic studies [28]. The criteria used to evaluate the quality of the cell culture studies were vague, resulting in four [44,45,46,47] cell culture studies considered to be high-quality (7–8) and two [21,40] cell culture studies considered to be of moderate quality (5–6). The QA scores for the individual studies are provided in the Appendix A.

### 3.3. RAGE Expression in Clinical PCa Samples

Eight clinical articles evaluated the expression of RAGE in PCa specimens compared with benign prostate samples mainly using immunohistochemistry (IHC) [36,37,38,39,40,41,42,43]. Due to study heterogeneity (*I*^2^ = 62.1%, Cochran’s *Q p* = 0.010), a random-effects model was performed. The meta-analysis of these studies confirmed that the OR was 11.3 (95% CI: 4.4–29.1), indicating an extremely strong likelihood for PCa to express RAGE compared to benign prostate tissues (Figure 2).

Funnel plots were used to visually assess publication bias (Appendix A), and while slightly asymmetrical, our search strategy attempted to minimize these effects by including unpublished (“grey”) literature. As such, we examined the studies for small-study effects using Harbord’s test and found no significant effect (*p* = 0.703; Appendix A). Furthermore, leave-one-out sensitivity analyses consistently demonstrated elevated ORs, ranging from 9.7–15.1 (Appendix A).

### 3.4. RAGE Expression in High- vs. Low-Grade PCa

Six clinical manuscripts additionally evaluated the expression of RAGE in high- vs. low-grade PCa specimens [24,36,38,39,40,43]. The data were reported as a grade and/or Gleason score; low-grade PCa was considered to be grade 3 and/or a Gleason score ≤ 7, and high-grade PCa was considered to be grade 4–5 and/or a Gleason score > 7. Due to two studies with low sample sizes yet high effect sizes [24,38], increased study heterogeneity was observed (*I*^2^ = 98.9%, Cochran’s *Q p* < 0.0001). The random-effects meta-analysis of this association produced an overall OR of 2.5 (95% CI: 1.8–3.4; Figure 3), indicating that greater RAGE expression is likely to be observed as PCa progresses.

### 3.5. Effect of RAGE on PCa Growth

Five of the six cell culture studies [40,44,45,46,47] evaluated the association between enhanced RAGE expression and PCa cell proliferation. Despite large variations in the methodologies used, the types of treatments performed, and the doses utilized, all five of the studies using human PCa cell lines reported that increased RAGE expression resulted in increased PCa cell proliferation.

Ishiguro et al. [40]. reported that the androgen-insensitive DU145 cell line expressed RAGE to a greater extent than androgen-sensitive LNCaP or androgen-insensitive PC-3 PCa cell lines, informing their decision to utilize the DU145 cell line to study the effects of RAGE expression on proliferation. They found that activating RAGE with an AGE ligand stimulated cell proliferation significantly. Similarly, Wu et al. [47]. showed that inhibiting the HMGB1/RAGE axis using verbascoside, derived from medicinal plants with antibacterial and anti-inflammatory properties at 0.1–10 µM, dose-dependently decreased RAGE expression and, consequently, cell proliferation in the DU145 and PC-3 cell lines. Both studies reported that simply modifying the activation status of RAGE had demonstrable effects on PCa cell proliferation.

Bao et al. [44]. found that treating with AGEs (1–400 µg/mL) increased cell proliferation in PC-3 cells. They also found that using RAGE siRNA to silence RAGE expression prior to AGE treatment removed this increase in proliferation back to control levels. This demonstrates a direct link between PCa proliferative potential and RAGE expression. This finding is further supported by Siddique et al. [46], who found that treating LNCaP cells with recombinant S100A4 (2 µg/mL), a RAGE ligand, significantly increased ^3^H thymidine uptake, indicating increased cell proliferation. RAGE siRNA was again confirmed to significantly reduce RAGE expression and subsequent cell proliferation. Elangovan et al. [45]. also reported that treating LNCaP and DU145 cells with recombinant HMGB1 increased cell proliferation in comparison to control cells. When pretreated with a shRNA plasmid targeting RAGE, RAGE expression in both cell lines decreased, and PCa cell proliferation decreased in comparison to control cells. These three studies show that when RAGE expression is silenced, PCa cell proliferation is significantly reduced as well.

An albatross plot (Figure 4A) was generated to integrate the data, and visual inspection of the plot provided an estimated standardized effect between RAGE expression and PCa growth. The effects given are not intended to be precise, as they only provide estimates of the magnitude of the effect of RAGE expression on PCa outcomes. The overall SMD range of 1.9 to 5.0 represents a strong overall positive effect. Two studies [40,45] showed moderate positive effects of RAGE on PCa proliferation, with an SMD of 1.9. The other three studies [44,46,47] demonstrated strong positive associations, with SMDs between 3.5 and 5.0, indicating an increase in PCa proliferation through the expression and activation of RAGE (combined Fisher’s *p* = 6.4 × 10^−6^). It is important to note that no studies reported a decrease in PCa proliferation with RAGE expression or activation, with the overall results demonstrating that RAGE expression and activation encourage PCa proliferation.

### 3.6. Effects of RAGEs on PCa Metastatic Potential

Three studies [21,40,47] evaluated the effects of modulating RAGE on PCa cell invasive and migratory features, or metastatic potential. Despite each study investigating different aspects of modulating RAGE, all three reported that increasing RAGE expression and activation significantly increased PCa invasion, migration, and the expression of key EMT markers.

Ishiguro et al. [40]. reported that stimulating RAGE in DU145 cells with AGE-BSA increased matrix metalloproteinases *mmp2* and *mmp9* expression and increased activated p44/p42 MAPK protein expression. MMP-2/-9 and activated p44/p42 MAPK are important for the process of invasion and metastasis and serve as critical EMT markers. Wu et al. [47]. reported that treating DU145 cells with verbascoside and an HMGB1-inhibitor (glycyrrhizae) decreased RAGE expression, decreased the protein expression of critical TGF-β pathway markers (α-SMA and TGF-β RI), and increased the expression of E-cadherin, Smad4, and Smad7. The TGF-β pathway has been linked to increased proliferation and EMT, with the Smad family acting as regulators of this pathway. Both DU145 and PC-3 cells were found to have decreased invasion and migration in wound healing and Transwell assays after treatment, demonstrating the effects of modulating RAGE signaling on metastatic potential.

Zhang et al. [21]. reported that treating PC-3 cells with recombinant HMGB1 significantly increased the mRNA levels of RAGE and dose-dependently increased cell migration and invasion. They also found that silencing RAGE or treating with anti-RAGE neutralizing antibodies significantly decreased the measured mRNA levels of EMT markers, such as N-cadherin, CTGF, and the MMP family, while increasing the mRNA levels of the epithelial junction proteins E-cadherin and vitamin D3 receptor. In addition, treatment with siRNA or anti-RAGE antibodies decreased the migratory and invasive potential of the PC-3 cells, demonstrating that direct interference with RAGE expression is associated with decreased PCa metastatic potential.

An albatross plot (Figure 4B) was generated to integrate the data, and visual inspection of the plot provided an estimated standardized effect between RAGE expression and PCa metastatic potential. The individual datapoints represent each individual reported dataset of invasion or migration from each study, resulting in ten points from three studies. The overall SMD range of 2.0 to 10.0 represents an increase in metastatic potential through the expression and activation of RAGE. One dataset [40] showed relatively low positive effects of RAGE on metastatic potential, with an SMD of 2.0. Four datasets [21] showed relatively moderate positive effects with an SMD of 4.0, and five datasets [21,47] showed strong positive effects with SMDs between 4.5 and 10.0, indicating an increase in metastatic potential through the expression and activation of RAGE (combined Fisher’s approached zero with *p* < 0.0001). It is important to note that no studies reported a decrease in PCa metastatic potential with RAGE expression or activation, suggesting that RAGE expression and activation encourages PCa EMT, migration, and invasion.

## 4. Discussion

In this systematic review and meta-analysis of RAGE in clinical PCa specimens, we demonstrated that there is a high prevalence of RAGE expression in PCa compared to benign prostate tissue and that RAGE can be used as a biomarker to differentiate between high- and low-grade PCa. We further expanded this systematic review to include preclinical studies that investigated the association between RAGE and pro-tumorigenic effects in PCa. While no consistent strategies currently exist to perform meta-analyses in preclinical studies that have used various unique methodologies, we adapted the guidelines from the WCRF/UoB recommendations [28] to perform a pseudo-meta-analysis of these data. Critically, the results of this analysis show consistently strong positive associations between RAGE and PCa growth and metastatic potential across all evaluated studies.

RAGE, the physiologic receptor for AGEs, has attracted significant attention since its discovery [48] due to its diverse ligand repertoire and involvement in several pathophysiological processes linked to inflammation, including cancer [37,49]. It was demonstrated that RAGE expression is directly tied to the malignant potential of PCa through different signaling mechanisms [36,43], including the activation of critical processes that promote drug resistance, stimulate angiogenesis, and enhance invasiveness [44,45,50]. Moreover, recent studies provided a positive association between RAGE, its ligands, such as AGEs and DAMPs, and neuroendocrine differentiation of PCa, which correlates with tumor grade, loss of androgen sensitivity, auto/paracrine activity, and poorer prognosis [51].

Our results demonstrate for the first time that RAGE expression is elevated in PCa, with an overall OR of 11.3 when compared to benign prostate tissue. Importantly, many studies have evaluated PCa in comparison to prostate tissues with BPH rather than normal prostate tissue. This may be highly significant, as BPH has been shown to be a strong predictor for developing PCa [52]. No current biomarkers exist to differentiate BPH from PCa [53], but increased RAGE expression in high- vs. low-grade PCa suggest there may be an association with RAGE in BPH as well as PCa. Our results show that high-grade PCa was found to be much more likely to express RAGE compared to low-grade cancers, suggesting that RAGE could be used as a biomarker to differentiate among different gradations of PCa.

Importantly, RAGE expression may be used for assessment when indolent cancers undergo a phenotypic switch to more aggressive, high-grade cancers. This is further evidenced by the role we identified in preclinical studies that RAGE plays in promoting PCa growth and metastatic potential. Cell culture studies consistently demonstrated that the expression and activation of RAGE is directly linked to PCa cell proliferative and migratory abilities, shown to occur mainly through the stimulation of the PI3K/Akt pathway and ultimately leading to the activation of oncoprotein NF-κB. HMGB1 expression in PCa was also investigated in several of the included studies, and HMGB1-specific activation of the RAGE axis was found to play a prominent role in the measured outcomes of PCa [21,40,41,43]. Because only two studies [41,43] compared the co-expression of RAGE and HMGB1, a meta-analysis was not conducted. As a DAMP, HMGB1 release by necrotic cells plays a role in priming immune cells to recognize dead or damaged tumor cells; however, this acute inflammatory effect by HMGB1 also results in sustained inflammation in the PCa microenvironment by RAGE ligand activity, ultimately promoting treatment resistance and tumor growth through RAGE activation [54,55].

Due to the high association identified between RAGE and PCa, RAGE expression could be used as a prognostic tool to monitor BPH and localized PCa. However, the risks and side effects associated with repeated biopsies to longitudinally monitor RAGE expression are numerous [2], and, as such, are not clinically feasible. To this end, our research group has developed a multimodal imaging platform to non-invasively quantify RAGE expression in tissues, with the probe having demonstrated consistent utility in imaging RAGE in PCa [24,25]. This platform has the potential to transform current diagnostic and therapeutic paradigms of PCa treatment by enabling clinicians to use medical imaging tools to non-invasively and longitudinally monitor BPH or localized PCa. Increases in RAGE expression over time would indicate PCa progression, providing a necessary criterion that will help determine clinical therapeutic response.

The strengths of this study include its novelty in identifying RAGE as a potent biomarker for PCa, with these associations remaining strong following a leave-one-out sensitivity analysis. The OR values remained above 9.0 in all scenarios, demonstrating a robust association between RAGE and PCa expression. The bias and sensitivity analyses were all supportive of the results, which lends confidence to the strength of the associations between RAGE expression and PCa. The limitations of this study include low study numbers and high intra-study heterogeneity due to the examination of RAGE expression in small numbers of PCa specimens. However, random-effects models were run to account for this heterogeneity when examining both the clinical and preclinical outcomes. Although preclinical studies demonstrated consistently positive and robust associations with RAGE and PCa growth and metastatic potential, albatross plots can only be used as an estimation of the overall effect and not the absolute effect. Despite this, the fact that all examined studies, albeit few in number, showed positive associations with the preclinical outcomes demonstrates the high potential for RAGE to serve as a biomarker that potently contribute to PCa tumorigenesis.

The results of this novel systematic review and meta-analysis will be clinically meaningful, as dietary modification may provide PCa patients with an adjuvant treatment to existing therapies. A dietary reduction of AGEs has been found to result in reduced RAGE expression and significant changes in the outcomes of other inflammatory-related diseases [56]. As such, the results of this meta-analysis demonstrating the effects of RAGE expression on PCa can be immediately translatable to clinical practice. Additional research efforts will be needed to understand how targeted anti-RAGE axis therapies may work to prevent PCa progression in both preclinical and clinical models, which our team is actively pursuing.

## 5. Conclusions

This systematic review and meta-analysis of clinical and preclinical studies demonstrated a robustly positive association between RAGE and PCa. The results of this study provide novel insight into a potential diagnostic and therapeutic biomarker for PCa and immediately actionable evidence for clinical recommendations for dietary modification in PCa.

## Figures and Tables

**Figure 1 cancers-15-04889-f001:**
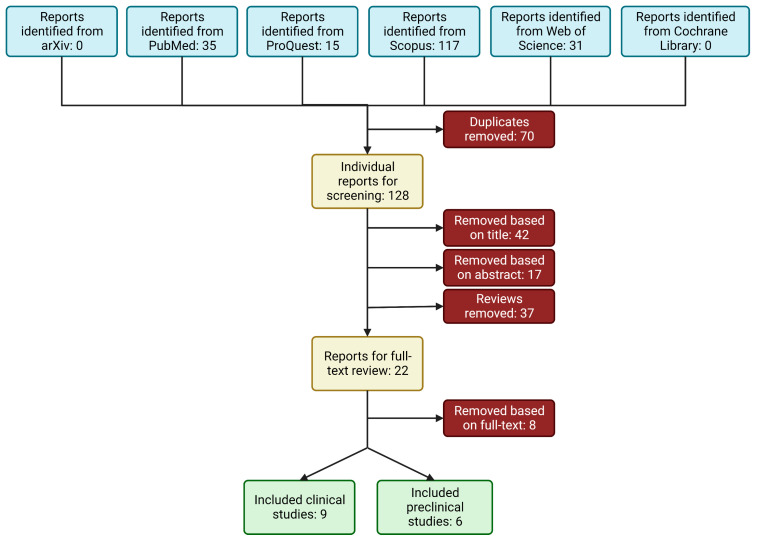
Schematic of workflow for identifying relevant articles.

**Figure 2 cancers-15-04889-f002:**
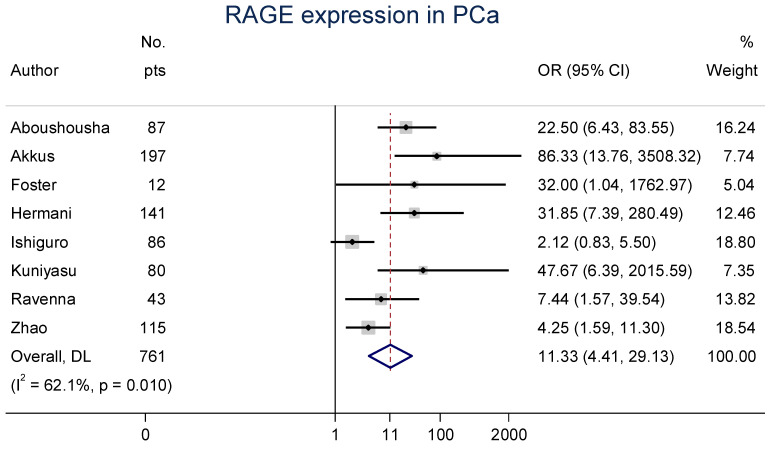
Forest plot of receptors for advanced glycation end-product (RAGE) expression in prostate cancer (PCa) [36,37,38,39,40,41,42,43]. These associations were indicated as odds ratio (OR) estimates with a corresponding 95% confidence interval (CI).

**Figure 3 cancers-15-04889-f003:**
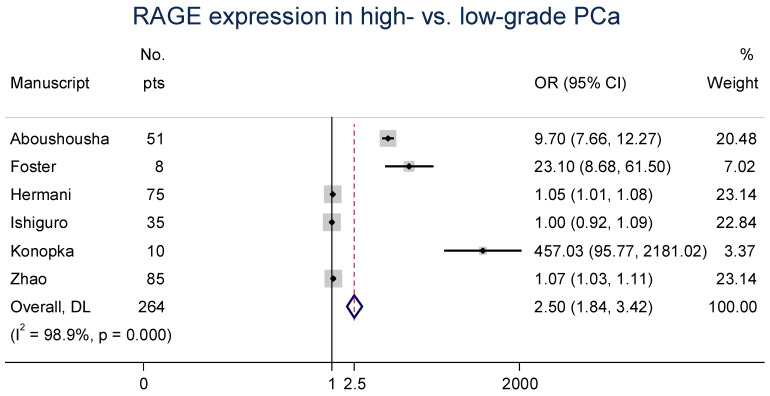
Forest plot of RAGE expression in high- vs. low-grade PCa [36,38,39,40,41,43]. These associations were indicated as OR estimates with the corresponding 95% CI.

**Figure 4 cancers-15-04889-f004:**
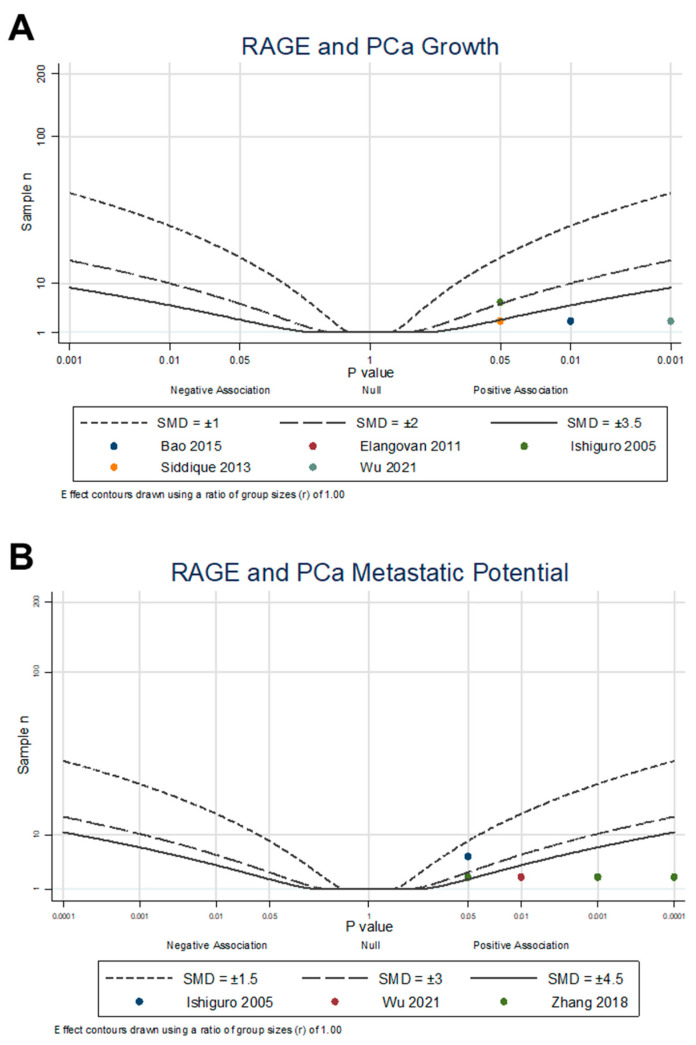
Albatross plot of cell culture studies measuring RAGE association and modulation with (**A**) PCa cell proliferation, measured by growth assays [40,44,45,46,47]; and (**B**) PCa metastatic potential, measured by invasiveness, migration, and EMT expression assays [21,40,47]. Each point represents a single study, with the effect estimate (represented as a *p*-value), plotted against the total given sample size (n) included within each study. Contour lines are standardized mean differences (SMD). Non-exact *p*-values reported were plotted as stated in the manuscript (e.g., if *p* < 0.05, plotted *p* = 0.05) as a conservative estimate.

**Table 1 cancers-15-04889-t001:** Characteristics of clinical studies included in the meta-analysis.

Author, Year	Control	Subjects (n)	Patient Characteristics ^1^	Method for RAGE	OR (95% CI) ^2^	*p*-Value ^3^
Aboushousha, 2019 [36]	BPH, prostatitis	PCa = 51 BPH = 20 prostatitis = 16	Treatment-naïve	IHC	22.5 (6.43–83.6)	**<0.0001**
Akkus, 2020 [37]	BPH	LPCa or MetPCa = 133 BPH = 64	Metastatic and localized, radical prostatectomy	IHC	86.3 (13.8–3508)	**<0.0001**
Foster, 2014 [38]	Healthy prostate	PCa = 26	Not listed	IHC	32 (1.04–1762)	**0.01**
Hermani, 2005 [39]	BPH, healthy prostate	PCa = 75 BPH = 56 Healthy prostate = 18	Radical prostatectomy	IHC	31.8 (7.39–280)	**<0.0001**
Ishiguro, 2005 [40]	BPH, healthy prostate	PCa = 43 BPH/healthy prostate = 43	Treatment-naïve and hormone refractory	RT-PCR	2.12 (0.826–5.50)	0.08
Konopka, 2020 [24]	none	PCa = 10	Radical prostatectomy	Western Blot	N/A	N/A
Kuniyasu, 2003 [41]	None	LPCa = 18 MetPCa = 22	Metastatic and non-metastatic, non-treatment-naïve	IHC	47.7 (6.39–2015)	**<0.0001**
Ravenna, 2009 [42]	Healthy prostate	PCa = 20	Not listed	IHC	7.44 (1.57–39.5)	**0.003**
Zhao, 2014 [43]	BPH	PCa = 85 BPH = 30	Metastatic and non-metastatic, treatment-naïve	IHC	4.25 (1.59–11.31)	**0.001**

^1^ Description of treatment status, sample source, or localized vs. metastatic PCa. ^2^ Odds ratio (OR) and 95% confidence interval (CI) calculated from case-control outcomes. ^3^ Bolded *p*-values indicate significance (*p* < 0.05). Abbreviations: benign prostatic hyperplasia (BPH), prostate cancer (PCa), localized prostate cancer (LPCa), metastatic prostate cancer (MetPCa), immunohistochemistry (IHC), real-time polymerase chain reaction (RT-PCR). N/A: not applicable.

**Table 2 cancers-15-04889-t002:** Characteristics of cell culture studies included in the systematic review and albatross plot analyses.

Author, Year	Cell Line(s)	Culture Conditions *	Treatment/Dose(s)	RAGE Expression	Cell Proliferation	Other Findings
Bao et al., 2015 [44]	PC-3	RPMI-1640, 10% FBS	AGES (1, 10, 100, 200, 400 µg/mL); RAGE siRNA	Confirmed lower RAGE expression after treatment with siRNA	Dose-dependent increase in proliferation with respect to AGE concentration (400 µg/mL; *p* < 0.05) and incubation time (48 h; *p* < 0.05). Silencing RAGE removed this effect.	
Elangovan et al., 2011 [45]	LNCaP, DU145	RPMI-1640, 10% FBS	shRAGE; rHMGB1 (1 µg/mL)	Significantly lowered RAGE expression in LNCaP and DU145 after shRAGE treatment (*p* < 0.01; *p* < 0.01)	Treating control with rHMGB1 increased cell proliferation, whereas down-regulating RAGE had a deleterious effect on proliferation in both cell lines with and without rHMGB1 (*p* < 0.05 for all groups).	Down-regulating RAGE increased Casp-3 and -8 expression (*p* < 0.05) and decreased PSA levels (*p* < 0.05).
Ishiguro et al., 2005 [40]	DU145	MEM, 10% FCS	AGE-BSA, 200 µg/mL	Higher RAGE expression in DU145 than PC-3 or LNCaP; study then used DU145	200 µg/mL AGE-BSA stimulated growth of DU145 compared to no treatment and BSA (*p* < 0.05).	200 µg/mL AGE/BSA increased avg number of invasive cells compared to BSA (*p* < 0.05). MMP-2, MMP-9, and activated phosphor-p44/p42 increased in AGE-BSA treatment group.
Siddique et al., 2013 [46]	LNCaP, PC-3	Not stated	RAGE siRNA (200 nM); rhS100A4 (2 µg/mL)	Confirmed lower RAGE expression after treatment with siRNA (100 nM, 200 nM)	rhS100A4 increased proliferation (*p* < 0.05), while silencing RAGE reduced it back to the control group’s level.	rhS100A4 increased NF-kB activity (*p* < 0.05) and silencing RAGE reduced back to control group levels.
Wu et al., 2021 [47]	DU145, PC-3	MEM, 10% FCS; RPMI-1640, 10% FBS	Verbascoside (0.1, 1, 10 µM)	Significantly lowered RAGE expression, dose-dependent (10 µM, *p* < 0.001)	Treating with verbascoside to inhibit RAGE decreased proliferation significantly (*p* < 0.001) in both cell lines.	Invasion was greatly reduced in DU-145 (*p* < 0.001) and PC-3 (*p* < 0.001) after verbascoside treatment, as was migration (*p* < 0.01, *p* < 0.001). EMT markers were also shown to significantly decrease after treatment.
Zhang et al., 2018 [21]	PC-3	DMEM, 10% FBS	rHMGB1 (1 µg/mL); anti-RAGE antibody (20 µg/mL); RAGE siRNA (5 nM)	Higher RAGE expression when treated with rHMGB1 (*p* < 0.0001)		siRAGE: Decreased migration compared to control and rHMGB1 groups (*p* < 0.01, *p* < 0.05) and decreased invasion in both as well (*p* < 0.0001, *p* < 0.001). Decreased EMT markers across the board as well. Antibody: Decreased migration compared to control and rHMGB1 (*p* < 0.01, *p* < 0.05) and decreased invasion (*p* < 0.001, *p* < 0.05). Decreased EMT markers.

* All studies reported standard incubator conditions (5% CO_2_, 37 °C) unless otherwise stated. Abbreviations: receptor for advanced glycation end-products (RAGE); small interfering (siRNA); advanced glycation end-products (AGEs); short hairpin RNA (shRNA); recombinant high-mobility group box 1 protein (rHMGB1); caspase protein (casp); prostate-specific antigen (PSA); bovine serum albumin (BSA); matrix metalloproteinase enzyme (MMP); recombinant human S100A4 (rhS100A4); nuclear factor kappa-light-chain-enhancer of activated B cells (NF-kB); epithelial-to-mesenchymal transition (EMT).

## Data Availability

This systematic review and meta-analysis was not registered under any program. The data generated in this study are available within the article and its Appendix A. Additional information pertaining to the literature search results and data extraction are available from the authors upon request.

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
