# Peer review of "RAGE as a Novel Biomarker for Prostate Cancer: A Systematic Review and Meta-Analysis"

_cancers, 2023, doi:10.3390/cancers15194889_

Round 1

Reviewer 1 Report

The topic of the paper "RAGEs and prostate cancer" is ineressant. 

However, apart from suboptimal English, the topic is not adequately addressed in the current version of the paper.

While the subject is well presented in the introduction, the method of collecting and evaluating the published papers on this topic, as well as the results and statistical method, are not presented in an understandable and structured manner. 

The discussion section contains incomprehensible sentences (as in paragraph 389) and repeated phrases.  

The whole text needs a moderate revision by a native English speaker.

Reviewer 2 Report

The authors of this review "RAGE as a novel biomarker for prostate cancer: a system review and meta-analysis" performed a excellent meta-analysis study correlating RAGE with PCa growth and potentially metastatic. Although it is a consultation work of data present in the literature, I believe that the authors have correlated the data extrapolated from various databases in a precise and rational way and that the results presented here are innovative and useful to the scientific community.

Author Response

The authors greatly appreciate the reviewer’s time and thoughtful critique of the manuscript presented. However, the authors do request the reviewer please re-visit the allocation of stars to reflect this kind review, as we believe there was an accidental misallocation to poorly review this manuscript. Thank you!

Reviewer 3 Report

In this Review, authors, after performing a systematic review and meta-analysis, conclude that the receptor for advanced glycation end products, RAGE, is a crucial biomarker of PCa progression, and can serve as an effective diagnostic target to differentiate between healthy prostate, low-grade PCa, and high-grade PCa.

The Review is original, very interesting, and has a potential translational value. Methodology is appropriate, Results are clearly reported and Discussion is well organized. 

Since this is a systematic Review and authors performed a meta-analysis, I would suggest to consider the following articles, that author might have not known. These are just suggestions that authors, of course, may freely consider or not.

- Pag. 2, lines 54-55: PMID: 20037200 and PMID: 19489685; 

- Pag. 2, lines 78-79: PMID: 34199263;

- Pag. 5, lines 210-211: PMID: 34638532. 

Author Response

The authors greatly appreciate the reviewer’s time and thoughtful critique of the manuscript presented and additionally appreciate the suggestion for the citations. Relevant citations have been updated and included in the introduction.

Round 2

Reviewer 1 Report

Although I could not identify the exact changes that have been made in this version of the manuscript compared to the old version of the paper, since the authors did not mark their edits in the paper, this version gives a better overall sense of the topic and is more understandable to the reader.